# Surgical Procedure Time and Mortality in Patients with Infective Endocarditis Caused by *Staphylococcus aureus* or *Streptococcus* Species

**DOI:** 10.3390/jcm11092538

**Published:** 2022-04-30

**Authors:** Gregor Paul, Laurin Ochs, Christopher Hohmann, Stephan Baldus, Guido Michels, Charlotte Meyer-Schwickerath, Gerd Fätkenheuer, Navid Mader, Thorsten Wahlers, Carolyn Weber, Norma Jung

**Affiliations:** 1Department I of Internal Medicine, Division of Infectious Diseases, University of Cologne, Kerpener Str. 62, 50937 Cologne, Germany; charlotte.meyer-schwickerath@uk-koeln.de (C.M.-S.); gerd.faetkenheuer@uk-koeln.de (G.F.); norma.jung@uk-koeln.de (N.J.); 2Department of Gastroenterology, Hepatology, Pneumology and Infectious Diseases, Katharinenhospital, Klinikum Stuttgart, Kriegsbergstraße 60, 70174 Stuttgart, Germany; 3Department III of Internal Medicine, University of Cologne, Kerpener Str. 62, 50937 Cologne, Germany; laurin.ochs@uk-koeln.de (L.O.); christopher.hohmann@uk-koeln.de (C.H.); stephan.baldus@uk-koeln.de (S.B.); 4Department of Acute and Emergency Care, St. Antonius Hospital Eschweiler, 52249 Eschweiler, Germany; guido.michels@sah-eschweiler.de; 5German Center for Infection Research (DZIF), Bonn-Cologne, 50937 Cologne, Germany; 6Department of Cardiothoracic Surgery, University of Cologne, Kerpener Str. 62, 50937 Cologne, Germany; navid.mader@uk-koeln.de (N.M.); thorsten.wahlers@uk-koeln.de (T.W.); carolyn.weber@uk-koeln.de (C.W.)

**Keywords:** endocarditis, surgery, staphylococcus aureus, streptococcus, survival

## Abstract

*Staphylococcus aureus* (SA) and *Streptococcus* species (SS) show different clinical manifestations in infective endocarditis (IE), but the impact on the complexity of surgical treatment remains unclear. All patients with surgically treated IE due to SA or SS between July 2013 and December 2016 were extracted from a prospectively collected, single-center registry. Data on patient characteristics, surgical procedures, and postprocedural outcomes were collected. SA-IE was more common with prosthetic valves (26.3% vs. 7.3%, *p* = 0.04), cardiac devices (14.3% vs. 0%, *p* = 0.03), previous cardiac surgery (28.6% vs. 9.8%, *p* = 0.03), intravenous drug abuse (14.3% vs. 0%, *p* = 0.03), and embolic events (57.1% vs. 26.8%, *p* = 0.007). Preoperative CRP was significantly higher in SA-IE (median 96.1 mg/L vs. 42.4 mg/L, *p* = 0.002). Otherwise, SS-IE affected more cusps/leaflets (mean 2.4 vs. 1.8, *p* = 0.03) and led to more valve dysfunction (83.8% vs. 54.3%, *p* = 0.007). Surgery times did not differ between the groups, though patients with SA spent more time in the intensive care unit (median 7 vs. 4.5 days, *p* = 0.04). Hospital mortality did not differ, but patients with SA-IE had unfavorable long-term survival (*p* = 0.001). Future studies need to be larger and focus on the mechanism behind the reduced long-term survival to mitigate the deleterious effect of SA in surgically treated patients with IE.

## 1. Introduction

Pre- and postoperative risk factors surrounding cardiovascular surgery have been extensively studied. Patient-related risk factors are age, female gender, extracardiac arteriopathy, chronic kidney failure, recent myocardial infarction, and congestive heart failure [1,2]. Some of these are modifiable, while others are not. Examples of modifiable factors are controlling diabetes and high blood pressure, quitting smoking, and correcting anemia. Nonmodifiable factors are age, sex, prior cardiac surgery, and the urgency of the procedure [3].

Surgery for active infective endocarditis (IE) is a known independent risk factor and is indicated in patients with resulting heart failure, uncontrolled infection, and for the prevention of systemic embolism [1,4]. However, the decision regarding the surgical treatment of IE is rather complex; therefore, several risk scores have been developed to predict the outcome after surgery for IE [5,6,7,8,9].

Only a minority of the scores include microorganisms to predict the result, such as the RISK-E score or the PALSULE score [5,8]. Additionally, only one study examined the influence of the detected microbiological organism in surgically treated patients with IE [10]. However, with the increasing number of prosthetic valves and intracardiac devices in the elderly, the risk of endocarditis increases, particularly in the case of *Staphylococcus aureus* (SA) [11]. Because of its high virulence, SA has been shown to be an independent risk factor for mortality in IE, but its influence on the outcome of surgery for IE is less well studied [12]. *Streptococcus* spp. (SS) and SA are two of the most common pathogens causing IE. For this reason, we decided to investigate and compare the influence of these two on surgical procedure times and mortality in patients with surgically treated IE [13].

## 2. Materials and Methods

### 2.1. Study Population

In this single-center cohort study, we prospectively included all patients that were evaluated for definite or possible IE, according to the revised Duke criteria [14], in connection with a native valve, prosthetic valve or cardiac implantable electronic device-related infective endocarditis. The information was collected between July 2013 and December 2016. Data on medical history, predisposing risk factors, demographics, laboratory results, microbiological organism, echocardiography, treatment (antibiotics or surgery), complications, intervention times, and outcome were recorded. All patients were cared for by a multidisciplinary endocarditis team consisting of at least a cardiologist, a cardiac surgeon, a microbiologist, and an infectious disease specialist.

IE was defined as noninvasive if it was confined to the cusps and leaflets and defined as invasive if it extended into the annulus and surrounding structures. Persistent bacteremia was defined as persistently positive blood cultures for more than 72 h after the initiation of an effective anti-infective therapy.

### 2.2. Statistics

Continuous data were expressed as mean with standard deviation (SD) in the case of normal distribution or median with interquartile range (IQR) for not normally distributed data. Categorical variables were reported as numbers (n) and percentages (%). Statistical differences between the groups were determined using the Student t-test (continuous variables with normal distribution), Mann–Whitney-U test (continuous variables without normal distribution), or the Chi-square test and the Fishers exact test (categorical variables). A Shapiro–Wilk test was performed to test for normality. The survival analysis was carried out using a Cox proportional hazard model. All demographics and clinical characteristics, including age and gender, with a *p*-value less than 0.05 were entered as covariates using a forward stepwise approach.

The flowchart was created using LibreOffice Draw (Version 7.2.4.1, The Document Foundation, Berlin, Germany). SPSS (SPSS 24, SPSS Inc., Chicago IL, USA) was used for statistical analysis. The reported ***p***-values are 2-tailed, with *p* < 0.05 considered to be statistically significant.

## 3. Results

### 3.1. Patient Characteristics, Comorbidities, and Risk Factors

Figure 1 shows a flow chart of the participants in this study. A total of 309 patients with a diagnosis of IE were prospectively registered during the study period. Of these, 155 patients (50.2%) were treated surgically.

This study focused on a comparison between patients with IE caused by *Staphylococcus aureus* (*n* = 42) and patients with IE caused by *Streptococcus* species (*n* = 41), all of whom were treated surgically. All detected SA were methicillin-sensitive SA (MSSA). SS consisted of 1 Lancefield group A Streptococcus (*S. pyogenes*), 3 group B Streptococci (2 *S. agalactiae*, 1 not further specified), 1 group C Streptococci (*S. dysgalactiae*), 8 group D Streptococci (8 *S. gallolyticus*), 25 are part of the viridans group streptococci (4 *S. mitis*, 4 *S. oralis*, 4 *S. sanguinis*, 3 *S. cristatus*, 3. *S. mutans*, 3 *S. gordonii*, 2 *S. salivarius*, 1 *S. anginosus*, 1 not further specified), and 2 *Streptococcus pneumoniae*. Patient characteristics and demographics are summarized in Table 1.

The median age was 57 years (interquartile range (IQR) 45.8–72.3) for SA and 58 years (IQR 50–70.5) for SS (*p* = 0.89). There were predominantly men in both groups (59.5% vs. 73.2%, *p* = 0.19). Comorbidities were mostly the same in both groups, although intravenous drug use was found exclusively in patients with SA (*n* = 6/42). Tricuspid valve endocarditis was also found only in patients with SA (*n* = 5/42). Diabetes was more common in patients with SA (26.2% vs. 10%, *p* = 0.06). Patients with SA endocarditis had undergone significantly more previous cardiac surgeries (28.6% vs. 9.8%, *p* = 0.03), as well as prosthetic valve endocarditis (26.2% vs. 7.3%, *p* = 0.04) and cardiac-device-related endocarditis (14.3% vs. 0%, *p* = 0.03). The cardiac device had been removed in all patients. In addition, patients with SA-associated IE had significantly more embolic events, especially to the brain and spleen prior to surgery (57.1% vs. 26.8%, *p* = 0.007), higher C-reactive protein levels (median 96.1 mg/dL vs. 42.4 mg/dL, *p* = 0.002), and a tendency for a higher frequency of extracardiac foci (25% vs. 7.7%, *p* = 0.07)

### 3.2. Surgical Indication, Procedure Times, and Postoperative Course

Table 2 shows the results for surgical procedures and postoperative outcomes.

The indication for surgery differed between the two groups (*p* = 0.009). The main indication for surgery was valvular dysfunction in both groups, but this was significantly more often the indication in patients with SS-related IE (54.3% vs. 83.8%, *p* = 0.007). One patient with persistent bacteremia in the SA-IE group had a relapse following a newly implanted valve and was therefore operated upon twice. Only data from the first surgery were used for calculations. The patient died eventually. More cusps or leaflets were affected in SS-IE as determined intraoperatively (mean 1.8 vs. 2.4, *p* = 0.03). Preoperative embolism was more often the reason for surgery in patients with SA-IE (37.1% vs. 13.5%, *p* = 0.02). Valve pathology was performed in twelve patients with SA-IE and eight patients with SS-IE. The pathological analysis affirmed IE in nine patients with SA-IE (75%) and seven patients with SS-IE (87.5%). Postoperatively, the incidence of stroke or intracerebral hemorrhage did not differ between the groups (10% vs. 11.8%, *p* = 1). Surgical procedure times or procedures performed were not different, but patients with SA-IE stayed longer in intensive care units (ICUs) (median 7 vs. 4.5 days, *p* = 0.04). None of the patients in either group died intraoperatively. While the time period of preoperative antibiotic therapy did not differ between groups, postoperative antibiotic therapy was significantly longer in patients with SA-IE (mean 24.9 days in SS-IE vs. 38.6 days in SA-IE, *p* = <0.001). In SA-IE, the most commonly used antibiotic was flucloxacillin (*n* = 31, 73.8%), followed by vancomycin (*n* = 11, 26.2%), daptomycin (*n* = 6, 14.3%), and cefazoline (*n* = 1, 2.4%). (More than one option was possible if the regimen was changed during the treatment course.) Combination therapy with rifampin was used in 19 patients (45.2%), with gentamicin in 14 patients (33.3%), and with fosfomycin in 4 patients (9.5%). In SS-IE, it was penicillin G (*n* = 32, 78%), vancomycin (*n* = 7, 17.1%), ampicillin (*n* = 4, 9.8%), and piperacillin/tazobactam (*n* = 2, 4.9%). Combination therapy with gentamicin was used in 22 patients (53.7%) and with rifampin in 3 patients (7.3%). The latter was used in cases of prolonged empiric therapy. Eight out of eleven patients with prosthetic valve endocarditis received combination therapy with either rifampin or fosfomycin. One patient discharged himself against medical advice and the empiric therapy was thereafter not adjusted. For one patient, no information was available on their postoperative antibiotic treatment regimen. Five out of six patients with cardiac-device-associated infective endocarditis either received adjunctive treatment with rifampin (*n* = 3) or fosfomycin (*n* = 2). Postoperative complications, such as postoperative renal failure or stroke, did not differ between groups. Cox proportional hazard regression with multivariable adjustment for age, gender, and preoperative embolism shows the statistically reduced survival of patients with SA-IE compared to SS-IE (adjusted hazard ratio 2.01, 95% CI 2.01–14.6, *p* = 0.001) (Figure 2).

Cox regression adjusted for age, sex, and embolism prior to surgery shows a significantly lower one-year survival rate in patients with surgically treated SA-IE compared to patients with SS-IE (adjusted hazard ratio 2.01, 95% CI 2.01–14.6, *p* = 0.001).

## 4. Discussion

We present a comprehensive comparison of patients with SS- and SE-IE, focusing on the complexity of the surgical treatment and postprocedural course.

This study shows that in patients with surgically treated IE, while SA-IE was more commonly associated with prosthetic valve and cardiac-device-related endocarditis, and consequently patients were more likely to have had prior cardiac surgery, the surgical times as well as postoperative complications (e.g., post-operative stroke or renal failure) did not differ from those of patients with SS-IE. In addition, patients with SA-IE had more embolic events prior to surgery, particularly in the brain, and higher levels of CRP. Survival was significantly lower in patients with SA-IE, mainly driven by long-term survival since in-hospital mortality was the same in both groups.

Our data are consistent with the published literature on SA-IE, which shows that prosthetic valves, cardiac devices, and intravenous drug use correlate with SA-IE [15]. Diabetes is also a well-known risk factor for SA-IE, but was not statistically associated in our study, which could be explained by the relatively small sample size [16]. It is also a known fact that SA-IE is associated with a higher rate of embolic events and higher CRP levels compared to SS-IE [17,18]. The good agreement of our results with the published literature strengthens the validity of our data.

To the best of our knowledge, no other study has examined the impact of microorganisms, particularly SA, on procedure times during surgery for IE. Our results demonstrate that SA did not affect procedure times such as total procedure time, bypass time, aortic clamp time, or reperfusion time. One explanation could be that the intraoperative invasiveness and incidence of abscesses were the same in both groups. Invasive disease has been shown to be a predictive factor for cardiopulmonary bypass time as well as aortic clamp time [19]. Interestingly, the intraoperative detection rate of valvular abscesses was nearly twice as high as the echocardiographic detection of abscesses. This affirms the importance of alternative diagnostic procedures, such as PET-CT, particularly in patients with prosthetic valve endocarditis.

Others have shown that the cardiac surgeon’s experience affects mean surgical procedure times [20]. In general, the greater the experience, the shorter the procedure times. Since we did not record individual surgeons, we cannot rule out that there was an imbalance in experienced surgeons between the groups. Studies have shown that previous cardiac surgery is associated with longer overall procedure times [21,22,23]. It cannot be excluded that operations that were expected to be more complex were assigned to more experienced surgeons, eventually compensating for longer surgical procedure times. However, patients with SA-IE had longer postoperative ICU stays and more ventilation hours. Though, the latter did not reach statistical significance.

Our results indicate that SS-IE affects more cusps or leaflets, as determined intraoperatively. We hypothesize that this is the reason why most patients with SS-IE had valvular dysfunction as their primary indication for surgery, while the prevention of (further) embolism was more often the indication in patients with SA-IE. We are not aware of any other study that has examined this fact, suggesting that this is a novel finding.

Williams et al. have shown that SA-IE is associated with a higher 30-day mortality when compared to SS-IE in surgically treated patients [10]. Han et al. were able to show that there was no difference in short-term survival between patients with SA-IE and other microorganisms, but that long-term survival was poorer in patients with SA-IE [24]. A similar pattern of results as in the study by Han et al. was obtained in our study. We see no difference in in-hospital mortality, but higher long-term mortality in patients with SA-IE. Survival curves between patients with SA-IE and SS-IE begin to diverge about 1 month after surgery, suggesting that any difference in short-term mortality could be mitigated by valvular heart surgery. However, it must be said that in this study, only patients who had undergone surgical intervention were examined. We cannot rule out that mortality rates would be different in patients that did not undergo surgical treatment. Studies that did not exclusively focus on patients undergoing surgical treatment showed higher mortality rates in patients with SA-IE [25,26]. The study by Pang et al. showed that SA-IE and prosthetic valve endocarditis are associated with reduced long-term survival after the surgical treatment of IE [27]. The reason for this phenomenon is unclear. Studies have shown that neurological complications of IE have a significant negative impact on the outcome, particularly in the long term [28,29,30]. As shown in our and other studies, cerebral embolism is more common in SA-IE, which may partially explain the poorer outcome in these patients.

The results of this study must be viewed in light of some limitations. First, the small sample size limits the statistical power to reveal small effects between groups. For instance, diabetes or hemodialysis are well-described risk factors for SA-IE and also occurred more frequently in patients with SA in our study, but the relatively small sample size allegedly led to the incorrect acceptance of the null hypothesis [31]. Additionally, procedure times were numerically longer in patients with SA-IE, but the sample size probably did not allow statistical distinction between groups. Secondly, SA and SS only represent two of many potential organisms of IE. Comparing different microorganisms can lead to different results. For instance, fungal IE is known to be associated with higher mortality than bacterial IE [10,32]. Therefore, it seems reasonable to assume that including more and different microorganisms in the analysis would have different effects on surgical procedure times and outcomes. Since SA and SS are two of the most commonly associated microbes in IE, we decided to focus our research question on them. Both limitations should be addressed in future research by assessing procedure times and outcomes in surgically treated IE on a larger dataset containing more microorganisms.

## 5. Conclusions

In summary, our work shows that *Staphylococcus aureus* endocarditis in surgically treated patients has no impact on surgical procedure times or in-hospital mortality compared to endocarditis caused by *Streptococcus* species. Consequently, the fear of more intraoperative complications in patients with SA-IE in comparison to SS-IE is not warranted and should not guide the decision-making process of the multidisciplinary heart team for or against surgery for IE. However, the postoperative stay in the intensive care unit is longer, and long-term survival is lower in patients with *S. aureus*-associated endocarditis. Future studies with larger cohorts and more microorganisms are needed to further investigate the influence of microorganisms on the outcome of surgically treated patients with infective endocarditis.

## Figures and Tables

**Figure 1 jcm-11-02538-f001:**
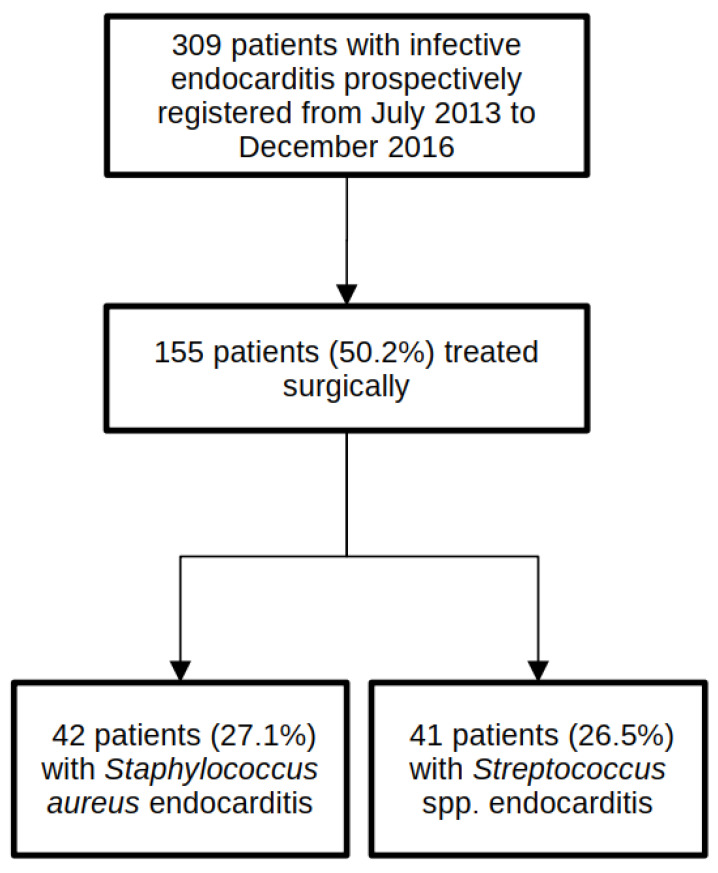
Flow diagram showing study population selection.

**Figure 2 jcm-11-02538-f002:**
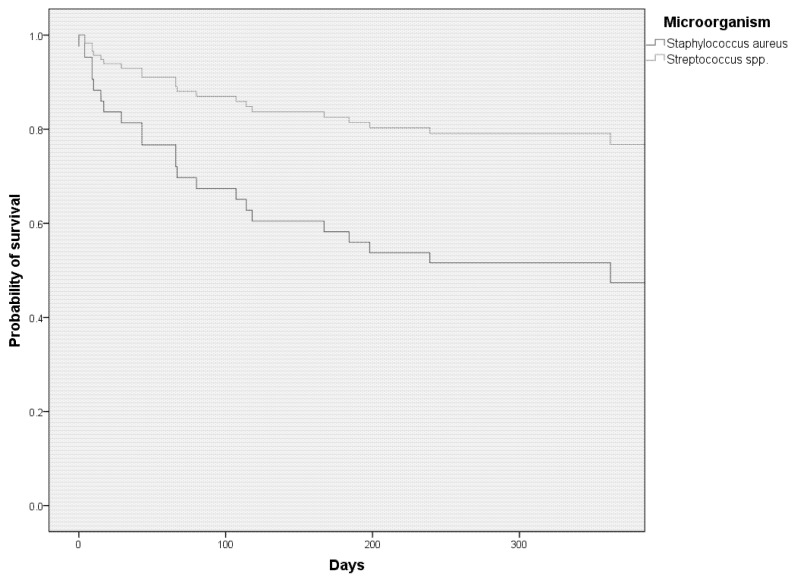
Cox proportional hazard regression showing impaired one-year survival in patients with surgically treated *Staphylococcus aureus* endocarditis compared to *Streptococcus* spp.

**Table 1 jcm-11-02538-t001:** Demographics and clinical characteristics of patients with infective endocarditis treated surgically and caused by *Staphylococcus aureus* or *Streptococcus* species.

	*Staphylococcus aureus* *n* = 42	*Streptococcus* Species *n* = 41	*p*-Value
	*n*/*n*_total_ (%)	*n*/*n*_total_ (%)	
**Demographics**			
Age, y; median (IQR)	57 (45.8–72.3)	58 (50–70.5)	0.89
Male sex	25/42 (59.5)	30/40 (73.2)	0.19
Diabetes	11/42 (26.2)	4/40 (10)	0.06
COPD	5/42 (11.9)	1/40 (2.5)	0.1
Carcinoma	1/42 (2.4)	1/40 (2.5)	0.97
IVDU	6/42 (14.3)	0/40	**0.03**
Chronic renal failure	13/42 (31)	10/41 (24.4)	0.63
Previous dialysis	7/41 (17.1)	2/34 (5.9)	0.17
New renal failure before surgery	10/42 (23.8)	6/41 (14.6)	0.29
Previous endocarditis	2/41 (4.9)	0/34	0.19
Previous cardiac surgery	12/42 (28.6)	4/41 (9.8)	**0.03**
Community-acquired IE	38/41 (92.7)	41/41 (100)	0.24
**Valve**			
Vegetation size, cm; mean (SD)	1.4 (0.95) *n* = 27	1.56 (0.64) *n* = 18	0.61
Prosthetic	11/42 (26.2)	3/41 (7.3)	**0.04**
Cardiac device (PM, ICD, or CRT)	6/42 (14.3)	0/40 (0)	**0.03**
Aortic	24/42 (57.1)	27/41 (65.9)	0.26
Mitral	21/42 (50)	19/41 (46.3)	1
Tricuspid	5/42 (11.9)	0/41 (0)	**0.03**
Pulmonary	0/42 (0)	3/41 (7.3)	0.07
Echocardiographic abscess	9/42 (21.4)	5/41 (12.2)	0.38
Intraoperative abscess	17/42 (40.5)	10/41 (24.4)	0.16
**Embolism and extracardiac focus prior surgery**			
Embolism (Yes/No)	24/42 (57.1)	11/41 (26.8)	**0.007**
No embolism	18/42 (42.9)	31/41 (75.6)	0.002
Cerebral	17/42 (40.5)	8/41 (19.5)	0.04
Spleen	6/42 (14.3)	1/41 (2.4)	0.05
Other	1/42 (2.4)	1/41 (2.4)	1
Extracardiac focus (e.g., osteomyelitis)	10/40 (25)	3/39 (7.7)	0.07
**Laboratory results**			
Preoperative WBC, T/µL; median (IQR)	10.6 (7.9–15)	10.3 (6.8–13.6)	0.67
Preoperative CRP, mg/L; median (IQR)	96.1 (32.8–177.3)	42.4 (11.25–76.2)	**0.002**

IQR, interquartile range; COPD, chronic obstructive pulmonary disease; IVDU, intravenous drug user; IE, infective endocarditis; PM, pacemaker; ICD, implantable cardioverter-defibrillator; CRT, cardiac resynchronization therapy; WBC, white blood cell count; CRP, C-reactive protein. Bold indicates a *p*-value below 0.05.

**Table 2 jcm-11-02538-t002:** Surgical procedures and postoperative outcome in patients with *Staphylococcus aureus* or *Streptococcus* spp. endocarditis.

	*Staphylococcus aureus* *n* = 42	*Streptococcus* Species *n* = 41	*p*-Value
	*n*/*n*_total_ (%)	*n*/*n*_total_ (%)	
Euroscore II, %; mean (SD)	8.5 (3.4) ***n*** = 41	7.9 (3.1) ***n*** = 38	0.5
Urgency of operation			0.77
Elective	16/42 (38.1)	16/41 (39)	
Urgent	19/42 (45.2)	16/41 (39)	
Emergency	7/42 (16.7)	9/41 (22)	
Primary indication for surgery			**0.03**
Embolism	13/35 (37.1)	5/37 (13.5)	0.02
Valvular dysfunction	19/35 (54.3)	31/37 (83.8)	0.007
Vegetation size	0/35	1/37 (2.7)	0.33
Abscess	2/35 (5.7)	0/35	0.14
Persistent bacteremia	1/35 (2.9)	0/35	0.3
Type of valve implanted			1
Biological	20/42 (47.6)	19/41 (46.3)	
Prosthetic	21/42 (50)	20/41 (48.8)	
Other	1/42 (2.4)	2/41 (4.9)	
Additional procedure			0.89
No additional procedure	23/42 (54.8)	25/49 (62.5)	
Endocarditis related	10/42 (23.8)	5/40 (12.5)	
Atrial appendage closure	2/42 (4.8)	3/40 (7.5)	
Patent foramen ovale	1/42 (2.4)	1/40 (2.5)	
Coronary artery graft bypass	4/42 (9.5)	4/40 (10)	
Reconstruction mitral or tricuspid valve	2/42 (4.8)	2/40 (5)	
Days antibiotics before surgery, days; mean (SD)	18.1 (17.7) N = 42	16.3 (17.2) N = 41	0.46
Days antibiotics after surgery, days; mean (SD)	38.6 (15.4) N = 32	24.9 (13) N = 34	**<0.001**
Valve culture			**0.47**
Positive	21/42 (50)	11/41 (26.8)	
Negative	21/42 (50)	28/41 (63.3)	
Not performed	0	2/41 (4.9)	
16 S rRNA PCR			**<0.001**
Positive	9/42 (21.4)	28/41 (68.3)	
Negative	7/42 (16.7)	7/41 (17.1)	
Not performed	26/42 (61.9)	6/41 (14.6)	
Operation time, minutes; median (IQR)	204 (148.5–256) N = 40	180 (154.25–216.75) N = 40	0.22
Bypass time, minutes; mean (SD)	114.5 (80.8–160) N = 40	107 (83–124) N = 39	0.39
Aortic clamp time, minutes, minutes; mean (SD)	80.5 (48–109.5) N = 42	71 (52.5–90.75) N = 40	0.68
Reperfusion time, minutes, minutes; mean (SD)	25 (20–36) N = 35	23.5 (18–35) N = 34	0.44
Invasive disease	14/42 (33.3)	9/41 (22)	0.33
Cusps/leaflets affected, N; mean (SD)	1.8 (1.16) N = 39	2.4 (1.16) N = 39	**0.03**
Postoperative pacemaker	4/42 (9.8)	3/34 (8.8)	1
Red blood packs, N; mean (SD)	3 (0.5–4.5) N = 41	2 (0–4) N = 41	0.22
Fresh frozen plasma, N; median (IQR)	0 (0–4) N = 41	0 (0–3.5) N = 41	0.42
Reoperation	5/42 (11.9)	1/41 (2.4)	0.2
Postoperative renal failure	12/42 (28.6)	8/41 (19.5)	0.44
Stroke/intracerebral bleeding after surgery	4/40 (10)	4/34 (11.8)	1
Ventilation hours, hours; median (IQR)	48 (13.2–225.9) N = 41	15.7 (10.5–80.9) N = 37	0.07
Days on intensive care, days; median (IQR)	7 (2–12) N = 41	4.5 (1.8–8.3) N = 38	**0.04**
Inhospital mortality	5/41 (12.2)	2/34 (5.9)	0.45

SD, standard deviation; rRNA, ribosomal RNA; PCR, polymerase chain reaction; IQR, interquartile range. Bold indicates a *p*-value below 0.05.

## Data Availability

The data that support the findings of this study are available on request from the corresponding author.

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
