# Peer review of "Surgical Procedure Time and Mortality in Patients with Infective Endocarditis Caused by Staphylococcus aureus or Streptococcus Species"

_jcm, 2022, doi:10.3390/jcm11092538_

Round 1

Reviewer 1 Report

The authors prospectively collected consecutive patients with definite and possible IE who underwent valve surgery between 2013-2016 at a center in Germany. The authors conducted the analytical cross-sectional study comparing surgical characteristics and mortality between SA-IE and SS-IE. The authors found that there were some differences in baseline characteristics and long-term mortality between groups. The study was well-conducted. These are some comments to clarify the overall understanding of the study.         

Major comments

  1. What is the hypothesis of this study that the authors hope to prove? Did the authors expect that there was a difference between the surgical characteristic of SS-IE and SA-IE? If so, what made the authors think that there might be a difference?
  2. How does the result - no difference in the characteristic of the surgical procedure between SA-IE and SS-IE, change/impact the current clinical practice?      
  3. The authors made a few conclusions from the findings, such as a higher rate of embolic events and higher CRP, and higher prosthetic valve/cardiac device/IVDU in the SA-IE. However, this may be misleading because the population of this study is the surgically indicated IE. This specific population was maybe sicker than usual IE patients. Therefore, it isn't easy to extrapolate this to the other SA-IE and SS-IE cohort.
  4. The critical information missing from the results is the details of post-surgical antibiotic therapy. Could the duration/type of antibiotic affect the long-term outcome?
  5. Other key findings that may alter the antibiotic management post-surgery include valve pathology, valve culture, 16S PCR of the valve. Did the author have this data? These factors may affect the outcome mainly, since the diagnosis includes both possible and definite IE (not purely definite).
  6. Additional information that may help strengthen this study is the rate of relapse of IE (with the same organism) after surgery, especially at 12 weeks timepoint.
  7. Was rifampin used in the case with prosthetic valve SA-IE? What was the management for the patient with persistent bacteremia?
  8. It may be helpful to report the adjusted HR rather than HR for survival since there are many confounders that could affect the outcome such as antibiotic use, severity of IE, comorbidities etc.

Minor comments

  1. What happened to the cardiac device? Did all the patients have the cardiac device removed during the surgery? If not, did the patient receive chronic antibiotic suppression therapy?
  2. Please double-check the number in this sentence – “25 are part of the viridans group streptococci (4 S. mitis, 4 S. oralis, 4 S. sanguinis, 3 S. cristatus, 3. S. mutans, 3 S. gordonii, 2 S. salivarius, 2 not further specified, 1 S. anginosus)”
  3. Which version of Duke criteria was used; the author should cite the criteria?

Reviewer 2 Report

It was an honor to review the manuscript Surgical procedure time and mortality in patients with infective
endocarditis caused by Staphylococcus aureus or Streptococcus species by paul et al.

Only a student t-test is used for analyzing continues variable. There is no mention of a kolmogorov-Smirnov test to asses normal distribution and subsequent use of either a Studen t-test or a  Mann-Whitney U test.

In table 1 vegetation size is named in mg/l. I assume this is in cm?

Round 2

Reviewer 1 Report

I thank the authors for revising this manuscript. The revision is satisfactory. No further comment.